# Frequency and Determinants of Olfactory Hallucinations in Parkinson’s Disease Patients

**DOI:** 10.3390/brainsci11070841

**Published:** 2021-06-24

**Authors:** Paolo Solla, Carla Masala, Ilenia Pinna, Tommaso Ercoli, Francesco Loy, Gianni Orofino, Laura Fadda, Giovanni Defazio

**Affiliations:** 1Department of Neurology, University of Sassari, Viale S. Pietro 10, 07100 Sassari, Italy; 2Department of Biomedical Sciences, University of Cagliari, SP 8 Cittadella Universitaria, 09042 Monserrato, Italy; ilenia.pinna.1994@gmail.com (I.P.); floy@unica.it (F.L.); 3Movement Disorders Center, Department of Neurology, University of Cagliari, SS 554 km 4.500, 09042 Cagliari, Italy; ercolitommaso@me.com (T.E.); dr.g.orofino@gmail.com (G.O.); fadda_laura@yahoo.it (L.F.); giovanni.defazio@unica.it (G.D.)

**Keywords:** Parkinson’s disease, olfactory dysfunctions, olfactory hallucinations, gustatory hallucinations

## Abstract

Background: Olfactory dysfunctions and hallucinations are considered common nonmotor symptoms in Parkinson’s disease (PD). Visual and auditory hallucinations are well-known; however, olfactory hallucinations (OHs) are not fully investigated. The aim of this study was to evaluate OHs in PD patients, and their correlation to motor impairment, cognitive abilities, visual and auditory hallucinations, and olfactory and gustatory function. Methods: A sample of 273 patients was enrolled: 141 PD patients (mean age ± SD: 70.1 ± 9.5 years) and 132 healthy controls (mean age ± SD: 69.4 ± 9.6 years). In all patients, the following parameters were evaluated: motor symptoms (UPDRS-III), olfactory function, cognitive abilities, and occurrence of OH, gustatory hallucinations (GHs), and visual/auditory hallucinations. Results: OHs were found only in PD patients with a percentage of 11.3%. Among PD patients with OHs, 2.8% also presented GHs. High significant frequencies of females, the presence of visual/auditory hallucinations, and a high mean UPDRS-III score were found in patients with OHs related to patients without them. Binary logistic regression evidenced the presence of visual/auditory hallucinations and sex as main variables predicting the presence of OHs. Conclusions: Our data indicated that OHs occur frequently in PD patients, especially in women, and often concomitant with visual and auditory hallucinations, without any association with olfactory impairment.

## 1. Introduction

Diagnosis of idiopathic Parkinson’s disease (PD) depends on the presence of a few motor symptoms, including bradykinesia, rigidity, tremor, and postural instability [1]. Nevertheless, PD is also characterized by the presence of a variety of nonmotor symptoms (NMSs) resulting from neurodegeneration-induced impairment in several systems such as olfaction, taste, vision, the gastric system, salivation, cardiovascular function, sleep, mood, and cognition [2]. 

Olfactory impairment is one of the most frequent nonmotor symptoms in PD [3,4,5,6]. It is usually reported in 65–90% of PD patients [7,8], may often precede the onset of motor symptoms, and may manifest as hyposmia/anosmia, parosmia (distorted perception of an odor), and phantosmia (perception of an odor in the absence of a relevant odor source), also indicated as olfactory hallucinations (OHs) [9,10].

While hyposmia/anosmia is widely characterized by means of validated tests (such as the University of Pennsylvania Smell Identification Test, UPSIT, and the Sniffin’ Sticks), OHs are not well investigated. A few previous studies, which focused on the frequency of OHs in PD [11,12,13,14,15], have provided inconsistent results, with estimates ranging between 2% and 10%. In addition, the relationship between OHs and quantitative impairment of the olfactory system in PD (hyposmia/anosmia) remains to be elucidated. Likewise, there are scant available data on the relationship between OHs and other relevant motor and nonmotor parkinsonian symptoms, also including other forms of hallucinations such as gustatory (GHs), visual, and auditory hallucinations [11].

The aim of the study was to evaluate OHs in PD patients, and their correlation to olfactory function, cognitive abilities, motor impairment, and gustatory, visual, and auditory hallucinations.

## 2. Materials and Methods

This cross-sectional study enrolled 273 subjects, 141 PD patients and 132 healthy controls, without any significant difference in sex (56 women and 85 men vs. 71 women and 61 men, *p* = 0.3) or age (70.2 ± 9.5 vs. 69.1 ± 9.6 years, *p* = 0.2). 

PD patients were consecutively recruited at the outpatient Movement Disorder Clinic of the University of Cagliari. PD was diagnosed according to the Gelb criteria [16]. Exclusion criteria were a history of head or neck trauma, stroke, atypical parkinsonism, dementia, psychiatric conditions, and chronic/acute rhinosinusitis. 

In PD patients, demographic information including age, sex, age at PD onset (the age at which the patient first observed initial motor symptoms), and current medications was collected by a structured clinical interview. The levodopa equivalent daily dose (LEDD) was computed as previously reported [17]. Motor impairment was assessed by the Unified PD Rating Scale (UPDRS) part III [18]. Cognitive status was evaluated by means of the Montreal Cognitive Assessment (MoCA), which measures cognitive abilities on different domains: visual-constructional skills, executive functions, attention and concentration, memory, language, conceptual thinking, calculations, and spatial orientation [19,20,21]. The total possible score was 30, with any score higher than 25 considered normal. Motor and cognitive assessments were performed in the on state.

Olfactory function was evaluated in PD patients and healthy control subjects by the Sniffin’ Sticks test (Burghart Messtechnik, Wedel, Germany), taking into consideration three parameters: odor threshold (OT), discrimination (OD), and identification (OI) [22,23,24,25,26,27]. All patients were instructed to drink only water 1 h before the test, and to avoid any smoking and scented product on the testing day. The Sniffin’ Sticks are pen-like odor-dispensing devices, and the complete procedure lasted 30–40 min [24,25]. Patients were blindfolded during the OT and OD task. First, OT was determined with 16 stepwise dilutions of n-butanol [28]. A three-alternative forced-choice task (3AFC) and the single-staircase technique were used [24,25,26,27]. Scores of OT ranged from 16 (patients who were able to detect the lowest concentration of n-butanol) to 1 (patients who were unable to detect the highest concentration). Second, OD was assessed using 16 trials. Three different pens were presented using the 3AFC task, two contained the same odor, and the third presented the target odorant. The OD score was calculated as the sum of the correct responses and ranged from 0 to 16 points [26,27]. Third, OI was evaluated using 16 common odors presented with four verbal descriptors in a multiple forced-choice format (three distractors and one target). The total score (threshold–discrimination–identification: TDI) was calculated: values > 30.5, ≤30.5, and ≤16.5 were considered normosmia, hyposmia, and functional anosmia, respectively [23,29]. PD patients and controls were also asked about OHs, and gustatory, visual, and auditory hallucinations observed in the last six months using questions about hallucinations from the Neuropsychiatric Inventory [30,31]. The occurrence of OHs was assessed by the question “Have you in the last year experienced so called phantom smells?” The question was answered on a 5-point Likert type scale, where 0 = “Never” and 4 = “Always”. Patients were grouped as phantosmic (1–4) and nonphantosmic (0 = never), as indicated by Sjölund et al. [32]. If patients showed a phantosmia, follow-up questions regarding the quality, frequency, and intensity of the phantosmia were asked. For each question, patients could answer with one of the four presented definitions, as previously reported by Landis et al. [33]. The sum of the four questions may range from 4 to 16 points [33].

Statistical analysis was performed by the SPSS software version 25 for Windows (IBM, Armonk, NY, USA). Data were expressed as mean and standard deviation (SD) unless otherwise indicated. Differences between groups were evaluated using one-way analyses of variance (ANOVA) with Bonferroni correction for multiple comparisons, or the chi-square (*χ^2^*) statistic with Yates correction, as appropriate.

In order to identify more promising factors for the binary logistic regression, bivariate correlations were performed between OHs vs. other hallucinations, sex, UPDRS, MoCA, LEDD, disease duration, and age using Spearman’s coefficient (r_s_). Binary logistic regression was used to determine which variable was the best significant predictor of OHs (the significance level was set at 0.05).

The study was approved by the local Ethics Committee (Prot. PG/2018/10157) and was performed according to the Declaration of Helsinki. Participants received an explanatory statement and gave their written informed consent to participate in the study. 

## 3. Results

In the parkinsonian group, the age at PD onset was 66.4 (SD: 10.2) years and the disease duration was 3.7 (SD: 3.2) years. The group mean UPDRS-III score during the on state was 20.7 (SD: 12.2), whereas the mean LEDD was 250 (SD: 237) and the mean MoCA score was 20.6 (SD: 5.5). 

Healthy controls did not show any olfactory, gustatory, visual, and auditory hallucinations (Table 1). Significant differences between PD patients and healthy controls were observed in olfactory and visual hallucinations (*p* = 0.0004 and 0.0001, respectively). In particular, in the PD group, OHs were reported by 16/141 (11.3%) subjects (Table 1), while visual hallucinations were reported by 25/141 (17.7%) patients. 

OHs were qualitatively heterogeneous: 3 patients (18.8%) described a perception of unpleasant odors, such as rotten eggs, garbage, or other noxious odors, while the remaining 13 (81.3%) reported pleasant smells such as flowers and fruits. Among PD patients with OHs, 4 (2.8%) also presented GHs. Visual and auditory hallucinations were reported by 25/141 (17.7%) and 6/141 (4.3%), respectively, in PD patients and 0/132 in healthy controls (Table 1). Nine patients shared OHs with visual hallucinations.

Assessing olfactory function by the Sniffin’ Sticks test yielded OT, OD, OI, and TDI scores that were significantly lower in PD patients as compared to the respective scores in healthy control subjects (Table 1). Among healthy controls, 44% (*n* = 58) showed normosmia, 54% (*n* = 71) presented hyposmia, and 2% (*n* = 3) were affected by functional anosmia.

Instead, among PD patients, 4% (*n* = 6) showed normosmia, 52% (*n* = 73) presented hyposmia, and 44% (*n* = 62) were affected by functional anosmia. Statistical differences between PD patients and healthy controls were observed in the percentage of anosmia (Yates *χ^2^* 39.7; *p* < 0.001) and normosmia (Yates *χ^2^* 35.9; *p* < 0.001).

In PD patients, women showed a high frequency of OHs (Table 2). In the same way, in patients with OHs, visual/auditory hallucinations increased, as well as UPDRS-III score.

The MoCA score was lower in the OHs group compared to those without OHs, but the difference did not reach the level of significance after Bonferroni correction. Patients with and without OHs were comparable for age, disease duration, LEDD, and TDI score (Table 2). Likewise, patients with OHs and those without OHs showed similar values of OT score (2.0 ± 1.4 vs. 2.7 ± 2.5; F_(1,139)_ = 1.290, *p* = 0.258), OD score (7.0 ± 2.5 vs. 7.5 ± 3.3, F_(1,139)_ = 0.448, *p* = 0.504), and OI score (7.0 ± 2.3 vs. 7.8 ± 3.5, F_(1,139)_ = 0.975, *p* = 0.325) on the Sniffin’ Sticks test.

Bivariate correlations were evaluated in order to assess correlations between olfactory hallucinations versus other hallucinations (visual and acoustic hallucinations), sex, UPDRS-III score, MoCA, LEDD, disease duration, and age (Table 3). Significant correlations were observed between olfactory hallucinations versus other hallucinations (r_s_ = 0.361, *p* < 0.01), versus sex (r_s_ = −0.258, *p* < 0.01), and versus UPDRS-III score (r_s_ = 0.199, *p* = 0.018). Instead, no significant correlations were found for MoCA, LEDD, disease duration, and age.

Finally, binary logistic regression analysis was performed in order to evaluate the main predictor variables on OHs (Table 4). The model showed significant effects (omnibus chi-square = 22.337, df = 8, *p* < 0.05). The model accounted for a range between 14.7% and 28.9% of the variance in OHs, with 98.4% of OHs successfully predicted. These data evidenced that the presence of visual/auditory hallucinations and female sex were the main variables predicting the presence of OHs.

## 4. Discussion

Assessing OHs in a large sample of nondemented PD patients in the early/intermediate stage of their clinical disease yielded a 11.3% frequency. A few investigations that previously dealt with this issue provided estimates ranging between 2% and 10% [11,12,13,14,15,34,35]. It is worth noting that our data on OHs replicated the results of the study by Bannier and Colleagues [14] (10%) that evaluated OHs not only using questionnaires but also analyzing quantitative olfactory function. Replicating the OHs frequency estimated in the higher part of the range of variability would suggest that the frequency of OHs may be underestimated. This may result from the general lack of unpleasantness of these experiences. Indeed, only a minority of PD patients reported situations of harmful or repulsive odors (cacosmia), while most of them (81.2%) described more often a pleasant smell.

In agreement with Bannier and Colleagues [14], we also observed no differences in OT, OD, and OI between PD patients with and without OHs. Thus, OHs in PD patients are probably not related to the quantitative olfactory dysfunction leading to hyposmia/anosmia, but rather to other mechanisms. In support of this view was the observed association between OHs and other forms of hallucinations such as gustatory, visual, and acoustic hallucinations, as indicated by Goetz et al. [36]. Interestingly, we found that GHs were present in 2.8% of PD patients, a frequency estimate that was higher than that released by the most recent comprehensive evaluation of these disperceptions [15]. 

Among the examined motor and nonmotor variables, visual/auditory hallucinations and female sex were the main variables predicting the presence of OHs. This observation adds to the body of evidence indicating that sex differences may play a key role in the development of NMSs in PD patients [34,37], and it also indicates peculiar features in olfactory dysfunction between the two sexes [34,37]. The lack of relationship between OHs and LEDD deserves further comments, bearing in mind previous studies that did not show significant differences between patients with or without current psychiatric symptoms in the cumulative dosage of levodopa [12]. Although the MoCA score tended to be lower in OH patients, the difference did not reach the level of significance after Bonferroni correction. However, in our study, we did not enroll patients with definite diagnoses of dementia. The strengths of our study include a consistent sample size resembling the demographic and clinical features of the general population of cases with early/intermediate clinical PD and the use of a psychophysical test such as the Sniffin’ Sticks test that consents the identification of a threshold, as well as of the discrimination and identification of smells. As the main limitations, this study was performed as a cross-sectional design, which enrolled patients and healthy controls from a single center.

## 5. Conclusions

Our findings indicated that OHs may be relatively common in PD patients. OHs were especially more frequent in female patients, often concomitant with other modalities of hallucinations, without a clear association with hyposmia/anosmia, disease duration, normal cognition/mild cognitive impairment, and LEDD. Interestingly, we identified a significant number of patients with GHs that were concomitant with OHs. These results emphasize the need for routine questioning to identify hallucinations in patients affected by PD and warrant further study to facilitate comprehension and treatment of these psychotic symptoms.

## Figures and Tables

**Table 1 brainsci-11-00841-t001:** Frequency of hallucinations and results of the Sniffin’ Stick test in patients with Parkinson’s disease and healthy control subjects.

	Parkinson’s Disease (*n*. 141)	Healthy Controls (*n*. 132)	*p* Value
Number of subjects with hallucinations			
Olfactory hallucinations (%)	16 (11.3%)	0 (0%)	0.0004
Gustatory hallucinations (%)	4 (2.8%)	0 (0%)	0.156
Visual hallucinations (%)	25 (17.7%)	0 (0%)	0.0001
Auditory hallucinations (%)	6 (4.3%)	0 (0%)	0.053
Sniffin’ Stick test			
Odor threshold score (mean ± SD)	2.8±2.9	5.4±3.6	0.0005
Odor discrimination score (mean ± SD)	7.8±3.4	12.4±2.4	0.0005
Odor identification score (mean ± SD)	7.5±3.2	11.1±2.5	0.0005
TDI score (mean ± SD)	17.8±7.3	28.7±6.1	0.0005

Legend: TDI = threshold–discrimination–identification scores. SD = standard deviation.

**Table 2 brainsci-11-00841-t002:** Clinical characteristics of PD subjects with or without olfactory hallucinations. Significance was set at the 0.0062 level after Bonferroni correction.

	Parkinsonian Patients with Olfactory Hallucinations (n.16)	Parkinsonian Patients without Olfactory Hallucinations (n. 125)	*p* Value
Number of women (%)	12 (75.0%)	44 (35.2%)	0.005
Age (mean years ± SD)	73.9 ± 8.1	69.7 ± 9.6	0.099
Disease duration (mean years ± SD)	3.9 ± 3.5	3.8 ± 3.2	0.912
Number of patients with visual/auditory hallucinations (%)	9 (56.3%)	20 (16%)	<0.0001
UPDRS-III score (mean ± SD)	29.5 ± 17.2	19.4 ± 11.0	0.0008
LEDD (mean ± SD)	260.6 ± 242.8	248.6 ± 236.9	0.849
TDI score (mean ± SD)	16.0 ± 4.8	18.1 ± 7.6	0.294
MoCA score (mean ± SD)	17.9 ± 6.1	21.0 ± 4.8	0.035

Legend: LEDD = levodopa equivalent daily dose; MoCA = Montreal Cognitive Assessment; TDI = threshold–discrimination–identification scores; SD = standard deviation; UPDRS-III = Unified PD Rating Scale part III.

**Table 3 brainsci-11-00841-t003:** Bivariate correlations between olfactory hallucinations versus other hallucinations (visual and acoustic), sex, Unified PD Rating Scale (UPDRS), Montreal Cognitive Assessment (MoCA), levodopa daily dose (LEDD), disease duration, and age.

	Olfactory Hallucinations	*p* Value
Other hallucinations	r_s_ = 0.361	***p* < 0.01**
Sex	r_s_ = −0.258	***p* < 0.01**
UPDRS-III score	r_s_ = 0.199	***p* = 0.018**
MoCA	r_s_ = −0.161	*p* = 0.057
LEDD	r_s_ = 0.012	*p* = 0.887
Disease duration	r_s_ = 0.004	*p* = 0.958
Age	r_s_ = 0.156	*p* = 0.065

Legend: LEDD = levodopa equivalent daily dose; MoCA = Montreal Cognitive Assessment; r_s_ = Spearman rank coefficient; UPDRS-III = Unified PD Rating Scale part III. Bold indicates significant level *p* < 0.05

**Table 4 brainsci-11-00841-t004:** Binary logistic regression models for clinical variables predicting presence of olfactory hallucinations in PD patients.

	B	S.E.	Wald	*p*	Exp(B)
Presence of visual/auditory hallucinations	1.543	0.649	5.658	**0.017**	4.680
Sex	−1.365	0.644	4.500	**0.034**	0.255
UPDRS-III score	0.030	0.022	1.822	0.177	1.030
MoCA score	−0.037	0.055	0.449	0.503	0.964

Legend: MoCA = Montreal Cognitive Assessment; S.E. = standard error; UPDRS-III = Unified PD Rating Scale part III. Bold indicates significant level *p* < 0.05.

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
