# Peer review of "Frequency and Determinants of Olfactory Hallucinations in Parkinson’s Disease Patients"

_brainsci, 2021, doi:10.3390/brainsci11070841_

Round 1

Reviewer 1 Report

In their manuscript “Frequency and determinants of olfactory hallucinations in Parkinson’s disease patients” (Manuscript ID: brainsci-1257079), Solla et al. describe that olfactory dysfunction and hallucinations are frequently observed in Parkinson's patients, with this non-motor dysfunction occurring more frequently in women. With their study, they are trying to close the gap in the correlation of Parkinson´s disease with impairments in olfactory function, something that has rarely been investigated in previous studies.

General comments:

In general, the manuscript by Solla et al. is well written and easy to follow. However, throughout, the manuscript should be revised with respect to English grammar and expression, ideally by a native speaker.

Moreover, the following issues must be addressed:

Major comments:

  1. In their introduction, as well as in the Materials/Methods part, the authors should go into more detail on olfactory hallucinations. For instance, how are olfactory hallucinations diagnosed, or can patients/probands diagnose themselves? How, if at all, are olfactory hallucinations classified, with respect to odor quality (or qualities?) and their descriptors, strength/severity, duration/persistency, and coincidence or non-coincidence with other olfactory stimuli? Here, the authors refer to a single reference (Landis & Burkhard, 2008) without giving any details – more background information is mandatory for non-specialist readers.

  1. While acknowledging the lack of investigations on olfactory hallucinations, this reviewer feels that the authors missed some of the literature available, such as:

Frei, K., and Truong, D.D. (2017). Hallucinations and the spectrum of psychosis in Parkinson's disease. J Neurol Sci 374, 56-62.

Goetz, C.G., Stebbins, G.T., and Ouyang, B. (2011). Visual plus nonvisual hallucinations in Parkinson's disease: development and evolution over 10 years. Mov Disord 26, 2196-2200.

The authors should mention/discuss these findings/reviews, at least the ones concerning olfactory halucinations.

  1. Line 93, Table 1: according to the references cited by the authors, all healthy control subjects in the present study were hyposmic (TDI score < 30.5). Explain!
  2. This reviewer assumes that the ‘Total score’ in the last line of Table 1 means the same as the ‘Sniffin’ stick test total score‘ in the last but one line of Table 2. Consequently, the authors should label these entries in both Tables identical as ‘TDI score’.
  3. What is the difference between the entry ‘UPDRS-III score’ in Table 2 and the entry ‘Unified Parkinson’s disease rating scale – part III’? What are the units?
  4. The binary logistic regression parameters given as column headers in Table 3 are unclear (what is ‘S.E.’ and what is ‘Wald’??). Authors must give the equation!

Minor comments:

Line 33: should read ‘...relies on the presence...’, or better: ‘...depends on the presence...’

Line 107: should read ‘...to participate in the study...’

Line 120: should read ‘...yielded OT, OD, OI, and TDI scores that were significantly lower in PD patients as compared to the respective scores in healty control subjects...’.

Line 140: In this sentence, which refers to Table 3, a significance level is given (p < 0.01) which is different from that given in Table 3, Line 146 (p < 0.05). Explain!

Line 160: should read ‘...are probably not related to the quantitative olfactory dysfunction leading to hyposmia/anosmia, but rather to other mechanisms.’.

Line 164: instead of ‘relieved by’, use either ‘released by’ or ‘reported by’.

Line 171: should read ‘...which did not show...’.

Line 175: should read ‘...enroll...’.

Line 180: The authors should revise the sentence here. In particular, ‘the setting of a cross-sectional analysis.’ does not constitute a complete sentence.

Line 186: should read ‘Interestingly, we identified a significant number of patients with GHs that were concomitant with OHs.’

Author Response

Reviewer 1:

Q 1. In their manuscript “Frequency and determinants of olfactory hallucinations in Parkinson’s disease patients” (Manuscript ID: brainsci-1257079), Solla et al. describe that olfactory dysfunction and hallucinations are frequently observed in Parkinson's patients, with this non-motor dysfunction occurring more frequently in women. With their study, they are trying to close the gap in the correlation of Parkinson´s disease with impairments in olfactory function, something that has rarely been investigated in previous studies.

General comments:

In general, the manuscript by Solla et al. is well written and easy to follow. However, throughout, the manuscript should be revised with respect to English grammar and expression, ideally by a native speaker.

Answer Q1: Authors thank the Reviewer for the observation and now the Manuscript has been revised by a native native English speaker. All revisions are indicated in red color on the manuscript.

Q 2. Major comments:

In their introduction, as well as in the Materials/Methods part, the authors should go into more detail on olfactory hallucinations. For instance, how are olfactory hallucinations diagnosed, or can patients/probands diagnose themselves? How, if at all, are olfactory hallucinations classified, with respect to odor quality (or qualities?) and their descriptors, strength/severity, duration/persistency, and coincidence or non-coincidence with other olfactory stimuli? Here, the authors refer to a single reference (Landis & Burkhard, 2008) without giving any details – more background information is mandatory for non-specialist readers.

Answer Q2: According the Reviewer’s observation, the Manuscripts in now changed into the following: “The occurrence of OHs was assessed by the question “Have you in the last year experienced so called phantom smells?”. The question was answered on a 5-point Likert type scale, where 0 = “Never” and 4 = “Always.” Patients were grouped as phantosmic (1–4) and nonphantosmic (0 = never) as indicated by Sjölund et al [31]. If patients showed a phantosmia follow-up questions regarding the quality, frequency and intensity of the phantosmia were asked. For each question patients could answer by one of the four presented definitions, as previously reported by Landis et al [32]. The sum of the four questions may range from 4 to 16 points [32].”

The qualitative olfactory evaluation is usually performed using self-reported questionnaires.

Q 3. While acknowledging the lack of investigations on olfactory hallucinations, this reviewer feels that the authors missed some of the literature available, such as:

Frei, K., and Truong, D.D. (2017). Hallucinations and the spectrum of psychosis in Parkinson's disease. J Neurol Sci 374, 56-62.

Goetz, C.G., Stebbins, G.T., and Ouyang, B. (2011). Visual plus nonvisual hallucinations in Parkinson's disease: development and evolution over 10 years. Mov Disord 26, 2196-2200.

The authors should mention/discuss these findings/reviews, at least the ones concerning olfactory halucinations.

Answer Q3: In line to the Reviewer’s observation these References are now included in the Discussion section.

Q4 a. Line 93, Table 1: according to the references cited by the authors, all healthy control subjects in the present study were hyposmic (TDI score < 30.5). Explain!

Answer Q4 a: Authors thank the Reviewer for the observation. As reported in previous studies (Hummel et al., 2007; Oleszkiewicz et al., 2019) an age-related decline in olfactory function indicated presbyosmia is usually observed in healthy controls. In particular, Hummel et al. (2007) indicated “Although an age-related loss of olfactory function is regularly observed, it

may not be inevitable.”.

In order to better explain this finding, we have included in the Results section the following description: “Among healthy controls, 44% (n = 58) showed normosmia, 54% (n = 71) presented hyposmia and 2% (n = 3) was affected by functional anosmia. Instead, among PD patients, 4% (n = 6) showed normosmia, 52% (n = 73) presented hyposmia and 44% (n = 62) was affected by functional anosmia. Statistical differences between PD patients and healthy controls were observed in the percentage of anosmia (Yates χ2 39.7; p < 0.001) and normosmia (Yates χ2 35.9; p < 0.001).”

Q4 b: This reviewer assumes that the ‘Total score’ in the last line of Table 1 means the same as the ‘Sniffin’ stick test total score‘ in the last but one line of Table 2. Consequently, the authors should label these entries in both Tables identical as ‘TDI score’.

Answer Q4b: According to the Reviewer’s suggestion, we have changed the Table 1 and Table 2 using the same definition.

Q 5. What is the difference between the entry ‘UPDRS-III score’ in Table 2 and the entry ‘Unified Parkinson’s disease rating scale – part III’? What are the units?

Answer Q5: We thank the Reviewer for this observation. There are no differences between UPDRS-III score’ in Table 2 and the Unified Parkinson’s disease rating scale – part III indicated in Table 1. Consequently, Authors decided to use UPDRS-III score in all tables and text description.

Q 6. The binary logistic regression parameters given as column headers in Table 3 are unclear (what is ‘S.E.’ and what is ‘Wald’??). Authors must give the equation!

Answer Q6: The logistic regression is the multivariate extension of a bivariate chi-square analysis. As reported by SPSS Manual (5th Edition): “Logistic regression analyses is a statistical procedure that can be used to predict category membership from a number of predictor variables.” Moreover, the Wald test is used to determine statistical significance for each of the independent variables. The S.E. rappresents the standard error around the coefficient for each variable. Binary logistic regression was performed using SPSS software version 25 for Windows (IBM, Armonk, N.Y., USA) which allows us to have different steps in the logistic regression model. Each step included different predictors, this is similar to blocking variables into groups and them into equation. By default, SPSS logistic regression is run in two steps:  the first step, called Step 0, this part of the output includes a “null model”, which is the model with no predictors and just the intercept, while the second step, called Step 1, includes predictors in the model. The Step 1 is the full model that we specified in the logistic regression command. 

The prediction equation is not reported by SPSS manual and guidelines. In addition, many other studies (e.g.: Junho et al., 2018; Simon-Gozalbo et al., 2020) did not report any information as regards equation to performe binary logistic regression using SPSS software. We reported the following References: 1) Junho, B. T., Kummer, A., Cardoso, F., Teixeira, A. L., & Rocha, N. P. (2018). Clinical Predictors of Excessive Daytime Sleepiness in Patients with Parkinson's Disease. Journal of Clinical Neurology (Seoul, Korea), 14(4), 530–536. https://doi.org/10.3988/jcn.2018.14.4.530. 2) Simon-Gozalbo A, Rodriguez-Blazquez C, Forjaz MJ and Martinez-Martin P (2020) Clinical Characterization of Parkinson’s Disease patients with cognitive impairment. Front. Neurol. 11:731. doi: 10.3389/fneur.2020.00731

Anyway, we indicate the following  general equation of binary logistic regression:

(please see attached file)

Minor comments:

Q 7. Line 33: should read ‘...relies on the presence...’, or better: ‘...depends on the presence...’

Answer Q7: Authors revised the Manuscript according the Reviewer’s suggestion and the sentence (lines 33-34)  is now changed into the following ”Diagnosis of idiopathic Parkinson's disease (PD) depends on the presence of a few motor symptoms including bradykinesia, rigidity, tremor and postural instability [1].”

Q 8. Line 107: should read ‘...to participate in the study...’.

Answer Q8: Authors revised the Manuscript according the Reviewer’s suggestion and the correction (line 112) is indicated in red color in the text.

Q 9. Line 120: should read ‘...yielded OT, OD, OI, and TDI scores that were significantly lower in PD patients as compared to the respective scores in healty control subjects...’.

Answer Q9: Authors revised the Manuscript according the Reviewer’s suggestion and the sentence (lines 129-131)  is now changed into the following ” Assessing olfactory function by the Sniffin’ test yielded OT, OD, OI, and TDI scores that were significantly lower in PD patients as compared to the respective scores in healthy control subjects (Table 1).”

Q 10. Line 140: In this sentence, which refers to Table 3, a significance level is given (p < 0.01) which is different from that given in Table 3, Line 146 (p < 0.05). Explain!

Answer Q9: Authors thank the Reviewer for this comment. The sentence is now changed in “The model showed significant effects (omnibus Chi-square = 22.337, df = 8, p < 0.05).”

Q 11. Line 160: should read ‘...are probably not related to the quantitative olfactory dysfunction leading to hyposmia/anosmia, but rather to other mechanisms.’

Answer Q11: Authors revised the sentence according the Reviewer’s suggestion: “Thus, OHs in PD patients are probably not related to the quantitative olfactory dysfunction leading to hyposmia/anosmia, but rather to other mechanisms.”

Q 12. Line 164: instead of ‘relieved by’, use either ‘released by’ or ‘reported by’.

Answer Q12. Authors revised the sentence according to the Reviewer’s suggestion and the correction is indicated on the text in red color.

Q 13. Line 171: should read ‘...which did not show...’.

Answer Q13. Authors thank the Reviewer and the correction is indicated in red color on the text.

Q 14. Line 175: should read ‘...enroll...’.

Answer Q14. Authors thank the Reviewer and the correction is indicated in red color on the text.

Q 15. Line 180: The authors should revise the sentence here. In particular, ‘the setting of a cross-sectional analysis.’ does not constitute a complete sentence.

Answer Q15. In line to the Reviewer’s comment the sentence is now changed into “As the main limitations, this study is performed as a cross-sectional design which en-rolled patients and healthy controls from a single center.”

Q 16. Line 186: should read ‘Interestingly, we identified a significant number of patients with GHs that were concomitant with OHs.’

Answer Q16. Authors thank the Reviewer and the correction is indicated in red color in the Conclusion section.

Reviewer 2 Report

Major comments: 
1. Demographics and results of clinical assessments are only presented for PD with vs without olfactory hallucinations with no details on the control performance. This is necessary to have a point of reference for the PD participants and should be included. 
2. A very coarse measure of cognition (MoCA) was used; this provides very limited information on cognitive state and is not very sensitive. Most PD patients with disease duration of 3.7 years, as in the presented study, would be expected to perform at ceiling. Indeed the performance in PD with a mean of 17.9 for PD with OH and 21.0 for PD without OH is surprisingly low.
Despite this, surprisingly, the authors state in line 129 that “However, most patients were in the normal range (not demented)”. This does not however fit with the presented mean/SD values and the authors cut-off of 25 as stated in the methods. 
Minor comments:
Logistic regression model: why were the specific variables (and only those) chosen for inclusion? The authors highlight throughout the manuscript the co-occurrence of gustatory hallucinations with OH but this was not included here. Why was logistic regression chosen for this rather than a simple metric of correlation such as Spearman rank coefficient? Particularly re motor severity (UPDRSIII) and MoCA this seems more reasonable.
Line 62: The authors state that they excluded those who " suffered from disorders that interfered with the correct assessment of clinical aspects of the disease". More detailed information on exclusion and exclusion criteria is needed as this is very vague. 
Line 155: “This may result from the general lack of unpleasantness of these despairs. Despairs should be changed to experiences.”

Author Response

Reviewer 2

Comments and Suggestions for Authors

Major comments:

  1. Demographics and results of clinical assessments are only presented for PD with vs without olfactory hallucinations with no details on the control performance. This is necessary to have a point of reference for the PD participants and should be included.

Answer 1: According to the Reviewer’s suggestion in the Results section is now included the following description: “Healthy controls did not show any olfactory, gustatory, visual and auditory hallucinations (Table 1). Significant differences between PD patients and healthy con-trols were observed in olfactory and visual hallucinations (p = 0.0004 and p = 0.0001, respectively). In particular, in the PD group OHs were reported by 16/141 (11.3%) sub-jects (Table 1), while visual hallucinations were reported by 25/141 (17.7%) patients.”

  1. A very coarse measure of cognition (MoCA) was used; this provides very limited information on cognitive state and is not very sensitive. Most PD patients with disease duration of 3.7 years, as in the presented study, would be expected to perform at ceiling. Indeed the performance in PD with a mean of 17.9 for PD with OH and 21.0 for PD without OH is surprisingly low.

Despite this, surprisingly, the authors state in line 129 that “However, most patients were in the normal range (not demented)”. This does not however fit with the presented mean/SD values and the authors cut-off of 25 as stated in the methods.

Answer 2: We thanks the Reviewer for the considerations and suggestions. We agree about the the fact that MoCA may provide only limited information on cognitive status. However, MoCA has been used consistently in PD studies, and is considered more appropriate regarding to other screening scales due to its psychometric properties (Scheffels et al. Concordance of Mini-Mental State Examination, Montreal Cognitive Assessment and Parkinson Neuropsychometric Dementia Assessment in the classification of cognitive performance in Parkinson's disease. J Neurol Sci. 2020 May 15;412:116735).

Consequently:

In the Methods section the sentence “Any score of 25 or less was considered a form of cognitive impairment (mild cognitive impairment was identified by the range 19 – 25)” is now deleted.

In the Results section the sentence “However, most patients were in the normal range (not emented)” is now deleted.

Moreover, our data are in line with those obtained by Yu and Colleagues (2020), where Authors reported: “We found that patients with an MMSE > 25 or a MoCA > 21 were less likely to have MCI.” In this study we obtained a mean value ± standard deviation of 21.0±4.8 of MoCA in Parkinsonian patients without olfactory hallucinations. 

However, contradictory data are reported as regards the use of MoCA scale for the evaluation of cognitive function in PD patients. Some studies (Chou et al., 2010; Dalrymple-Alford et al., 2010; Hu et al., 2014; Kandiah et al., 2014; Biundo et al., 2016; Scheffels et al., 2020) indicated a better sensitivity of MoCA than MMSE in PD patients’ cognitive evaluations since it contains attention and executive items. Instead, Lessig and Colleagues (2012) showed a better sensitivity of MMSE.

Please see the following References:

-Chou et al., (2010) A recommended scale for cognitive screening in clinical trials of Parkinson’s disease. Mov Disord 25:2501–2507

- Hu et al., (2014) Predictors of cognitive impairment in an early stage Parkinson’s disease cohort. Mov Disord 29:351–359

- Kandiah et al., (2014) Montreal Cognitive Assessment for the screening and prediction of cognitive decline in early Parkinson’s disease. Parkinsonism Relat Disord 20:1145–1148

- Biundo et al., (2014) Cognitive profiling of Parkinson disease patients with mild cognitive impairment and dementia. Parkinsonism Relat Disord 20:394–399

- Scheffels et al., (2020) Concordance of Mini-Mental State Examination, Montreal Cognitive

Assessment and Parkinson Neuropsychometric Dementia Assessment in the classification of cognitive performance in Parkinson's disease. J Neurol Sci 412, 116735.

- Lessig et al., (2012) Changes on briefcognitive instruments over time in Parkinson’s disease. Mov Disord 27:1125–1128

Minor comments:

  1. Logistic regression model: why were the specific variables (and only those) chosen for inclusion? The authors highlight throughout the manuscript the co-occurrence of gustatory hallucinations with OH but this was not included here. Why was logistic regression chosen for this rather than a simple metric of correlation such as Spearman rank coefficient? Particularly re motor severity (UPDRSIII) and MoCA this seems more reasonable.

Answer 3: Authors thank the Reviewer for this observation. In our study, first were carried out statistical differences using one-way ANOVAs and post-hoc with Bonferroni’s multiple comparison test between PD patients vs healthy controls and between PD patients with OHs vs those without OHs.

In line to the Reviewer’s observation, were now performed bivariate correlations using Spearman’s coefficient (rs) in order to assess correlations between olfactory hallucination versus other hallucination (visual and acoustic hallucinations), sex, UPDRS-III score, MoCA, LEDD, disease duration and age.

Finally, the binary logistic regression analysis was performed, in order to evaluate the main predictor variables on OHs.

Consequently, Authors included the following sentences:

-In Materials and Methods: “In order to identify more promising factors for the binary logistic regression, were performed bivariate correlations between olfactory hallucinations vs other hallucinations, sex, UPDRS, MoCA, LEDD, disease duration and age using Spearman’s coefficient (rs).”

-In Results section: “Bivariate correlations were evaluated in order to assess correlations between olfactory hallucination versus other hallucination (visual and acoustic hallucinations), sex, UPDRS-III score, MoCA, LEDD, disease duration and age (Table 3). Significant correlations were observed between olfactory hallucinations versus other hallucinations (rs= 0.361, p<0.01), versus sex (rs = -0.258, p < 0.01) and versus UPDRS-III score (rs = 0.199, p = 0.018). Instead, no significant correlations were found for MoCA, LEDD, disease duration and age”

  1. Line 62: The authors state that they excluded those who " suffered from disorders that interfered with the correct assessment of clinical aspects of the disease". More detailed information on exclusion and exclusion criteria is needed as this is very vague.

Answer 4. Authors thank the Reviewer for this suggestion, this sentence is now included (Line 61-62) in the Manuscript “Exclusion criteria were a history of head or neck trauma, stroke, atypical Parkinsonism, dementia, psychiatric conditions, and chronic/acute rhinosinusitis.”

  1. Line 155: “This may result from the general lack of unpleasantness of these despairs. Despairs should be changed to experiences.”

Answer 5. The sentence is now changed into: “This may result from the general lack of unpleasantness of these experiences.

Reviewer 3 Report

This is an interesting, highly relevant addition to this area of research as the focus is almost entirely on visual, and to a lesser extent auditory hallucinations. Raising awareness and understanding the correlates of olfactory hallucinations is important. 

Apologies, but the language and style interfere with the reading to some extent, and this needs to be addressed before publication. 

For example Line 33 should be rephrased’ Diagnostic criteria for Parkinson’s disease (PD) require xxxxx'

I would also advise use of less abbreviations eg say olfactory hallucinations and gustatory hallucinations 

Author Response

Rewiewer 3

Comments and Suggestions for Authors

This is an interesting, highly relevant addition to this area of research as the focus is almost entirely on visual, and to a lesser extent auditory hallucinations. Raising awareness and understanding the correlates of olfactory hallucinations is important.

Q1: Apologies, but the language and style interfere with the reading to some extent, and this needs to be addressed before publication.

Answer Q1: Authors thank the Reviewer for the observation and now the Manuscript has been revised by a native native English speaker. All revisions are indicated in red color on the manuscript.

Q2: For example Line 33 should be rephrased’ Diagnostic criteria for Parkinson’s disease (PD) require xxxxx'

Answer Q2: According to the Reviewer’s comments the sentence is now revised in “Diagnosis of idiopathic Parkinson's disease (PD) depends on the presence of a few motor symptoms including bradykinesia, rigidity, tremor and postural instability [1].

Q3. I would also advise use of less abbreviations eg say olfactory hallucinations and gustatory hallucinations.

Answer Q3: Authors tried to reduce the number of abbreviations in the test as much as possible.
